# DiCE: The Infinitely Differentiable Monte Carlo Estimator

**Jakob Foerster, Greg Farquhar, Tim Rocktäschel, Shimon Whiteson**
University of Oxford / Correspondence to: `jakob.foerster@cs.ox.ac.uk`

**Maruan Al-Shedivat & Eric P. Xing**
Carnegie Mellon University

## Abstract

The score function estimator is widely used for estimating gradients of stochastic objectives in Stochastic Computation Graphs (SCG), *e.g.*, in reinforcement learning and meta-learning. While deriving the first order gradient estimators by differentiating a *surrogate loss* (SL) objective is computationally and conceptually simple, using the same approach for higher order gradients is more challenging. Firstly, analytically deriving and implementing such estimators is laborious and not compliant with automatic differentiation. Secondly, repeatedly applying SL to construct new objectives for each order gradient involves increasingly cumbersome graph manipulations. Lastly, to match the first order gradient under differentiation, SL treats part of the cost as a fixed sample, which we show leads to missing and wrong terms for higher order gradient estimators. To address all these shortcomings in a unified way, we introduce DiCE, which provides a single objective that can be differentiated repeatedly, generating correct gradient estimators of any order in SCGs. Unlike SL, DiCE relies on automatic differentiation for performing the requisite graph manipulations. We verify the correctness of DiCE both through a proof and through numerical evaluation of the DiCE gradient estimates. We also use DiCE to propose and evaluate a novel approach for multi-agent learning. Our code is available at `https://goo.gl/xkkGxN`.

## 1 Summary

The score function trick is used to produce Monte Carlo estimates of gradients in settings with non-differentiable objectives, *e.g.*, in meta-learning and reinforcement learning. Estimating the first order gradients is computationally and conceptually simple. While the gradient estimators can be directly defined, it is often more convenient to define an objective whose derivative is the gradient estimator and let the powerful automatic-differentiation (auto-diff) toolbox as implemented in deep learning libraries do the work for you. This is the method used by the *surrogate loss* (SL) approach (Schulman et al., 2015a), which provides a recipe for building a surrogate objective from a *stochastic computation graph* (SCG). When differentiated, the SL produces an estimator for the first order gradient of the original objective.

Unfortunately, the first order gradient estimators mentioned above are fundamentally ill-suited for calculating higher order derivatives via auto-diff. Due to the dependency on the sampling distribution, higher order gradient estimators require repeated application of the score function trick. Simply differentiating the first order estimator again, as was for example done by Finn et al. (2017), leads to missing terms, as shown by Al-Shedivat et al. (2017) and Stadie et al. (2018).

To obtain higher order score function gradient estimators, there are currently two unsatisfactory options. The first is to analytically derive and implement the estimators. However, this is laborious, error prone, and does not comply with the auto-diff paradigm. The second is to repeatedly apply the SL approach to construct new objectives for each further gradient estimate. However, constructing each of these new objectives involves increasingly complex graph manipulations, defeating the appeal of using a differentiable surrogate loss, as illustrated in Appendix B.

Moreover, to match the first order gradient after a single differentiation, the SL treats part of the cost as a fixed sample, severing the dependency on the parameters. In Appendix C.2 we show that this yields missing and incorrect terms in higher order gradient estimators. We believe that these difficulties have limited the usage and exploration of higher order methods in reinforcement learning tasks and other application areas that may be formulated as SCGs.

Therefore, we propose a novel technique, the *Infinitely **Di**fferentiable Monte-**C**arlo **E**stimator* (DICE), to address all these shortcomings. DICE constructs a single objective, $\mathcal{L}_{\boxdot}$, that evaluates to an estimate of the original objective, but can also be differentiated repeatedly to obtain correct gradient estimators of any order. Unlike the SL approach, DICE relies on auto-diff as implemented for instance in TensorFlow (Abadi et al., 2016) or PyTorch (Paszke et al., 2017) to automatically perform the complex graph manipulations required for these higher order gradient estimators.

DICE uses a novel operator, MAGICBOX($\boxdot$), which wraps around the set of those stochastic nodes $\mathcal{W}_c$ that influence each of the original losses, $c$, in an SCG. Upon differentiation, this operator generates the correct gradients associated with the sampling distribution:

$$\nabla_\theta \boxdot(\mathcal{W}_c) = \boxdot(\mathcal{W}_c) \nabla_\theta \sum_{w \in \mathcal{W}_c} \log(p(w;\theta)),$$

while returning 1 when evaluated: $\boxdot(\mathcal{W}) \rightarrowtail 1$. MAGICBOX can easily be implemented in standard deep learning libraries as follows:

$$\boxdot(\mathcal{W}) = \exp\left(\tau - \bot(\tau)\right),$$
$$\tau = \sum_{w \in \mathcal{W}} \log(p(w;\theta)),$$

where $\bot$ is an operator (*e.g.* 'stop_gradient' or 'detach') that sets the gradient of the operand to zero, so $\nabla_x \bot(x) = 0$ . Using $\boxdot$, we finally define the DICE objective, $\mathcal{L}_{\boxdot}$, that fulfils all requirements from above:

$$\mathcal{L}_{\boxdot} = \sum_{c \in \mathcal{C}} \boxdot(\mathcal{W}_c) c + \sum_{w \in \mathcal{S}} (1 - \boxdot(\{w\})) b_w.$$

Here the baseline $b_w$ is a design choice and can be any function of nodes not influenced by $w$, further details are provided in Appendix D.2.

We verify the correctness of DICE both through a proof, in Appendix D and through numerical evaluation of the DICE gradient estimates in Appendix E. To demonstrate the utility of DICE, in Section 2, we also propose and evaluate a novel approach for learning with opponent learning awareness (Foerster et al., 2018). We also open-source our code in TensorFlow. We hope this powerful and convenient novel objective will unlock further exploration and adoption of higher order learning methods in meta-learning, reinforcement learning, and other applications of SCGs.

## 2 EXPERIMENTS

While the main contribution of this paper is to provide a novel general approach for any order gradient estimation in SCGs, we also provide a proof-of-concept empirical evaluation for a set of case studies, carried out on the *iterated prisoner's dilemma* (IPD). In IPD, two agents iteratively play matrix games with two possible actions: (C)ooperate and (D)efect. The possible outcomes of each game are DD, DC, CD, CC with the corresponding first agent payoffs, -2, 0, -3, -1, respectively. This setting is useful because (1) it has a nontrivial but analytically calculable value function, allowing for verification of gradient estimates, and (2) differentiating through the learning steps of other agents in multi-agent RL is a highly relevant application of higher order policy gradient estimators in RL (Foerster et al., 2018).

**DICE for multi-agent RL.** In *learning with opponent-learning awareness* (LOLA), Foerster et al. (2018) show that agents which differentiate through the learning step of their opponent converge to Nash equilibria with higher social welfare in the IPD.

Since the standard policy gradient learning step for one agent has no dependency on the parameters of the other agent (which it treats as part of the environment), LOLA relies on a Taylor expansion of the expected return in combination with an analytical derivation of the second order gradients to be able to differentiate through the expected return after the opponent's learning step.

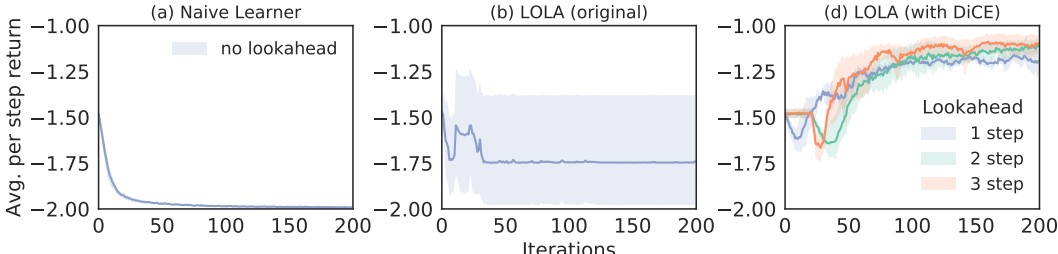

Figure 1: Joint average per step returns for different training methods. (a) Agents naively optimize expected returns w.r.t. their policy parameters only, without lookahead steps. (b) The original LOLA algorithm (Foerster et al., 2018) that uses gradient corrections. (c) LOLA-DICE with lookahead of up to 3 gradient steps. Shaded areas represent the 95% confidence intervals based on 5 runs. All agents used batches of size 64, which is more than 60 times smaller than the size required in the original LOLA paper.

Here, we take a more direct approach, made possible by DICE. Let $\pi_{\theta_1}$ be the policy of the LOLA agent and let $\pi_{\theta_2}$ be the policy of its opponent and vice versa. Assuming that the opponent learns using policy gradients, LOLA-DICE agents learn by directly optimising the following stochastic objective w.r.t. $\theta_1$:

$$\mathcal{L}^1(\theta_1, \theta_2)_{\text{LOLA}} = \mathbb{E}_{\pi_{\theta_1}, \pi_{\theta_2 + \Delta\theta_2(\theta_1, \theta_2)}} \left[ \mathcal{L}^1 \right], \text{where}$$
$$\Delta\theta_2(\theta_1, \theta_2) = \alpha \nabla_{\theta_2} \mathbb{E}_{\pi_{\theta_1}, \pi_{\theta_2}} \left[ \mathcal{L}^2 \right], \tag{2.1}$$

Here $\alpha$ is a scalar step size and $\mathcal{L}^i = \sum_{t=0}^{T} \gamma^t r_t^i$ is the sum of discounted returns for agent $i$.

To evaluate these terms directly, our variant of LOLA unrolls the learning process of the opponent, which is functionally similar to model-agnostic meta-learning (MAML, Finn et al., 2017). In the MAML formulation, the gradient update of the opponent, $\Delta\theta_2$, corresponds to the inner loop (typically training objective) and the gradient update of the agent itself to the outer loop (typically test objective).

Using the following DICE-objective to estimate gradient steps for agent $i$, we are able preserve all dependencies:

$$\mathcal{L}^i_{\boxdot(\theta_1, \theta_2)} = \sum_t \boxdot \left( \left\{ a_{j \in \{1,2\}}^{t' \leq t} \right\} \right) \gamma^t r_t^i, \tag{2.2}$$

where $\left\{ a_{j \in \{1,2\}}^{t' \leq t} \right\}$ is the set of all actions taken by both agents up to time $t$. We note that for computational reasons, we cache the $\Delta\theta_i$ of the inner loop when unrolling the outer loop policies in order to avoid recalculating them at every time step.

Importantly, using DICE, differentiating through $\Delta\theta_2$ produces the correct higher order gradients, which is vital for LOLA to function. In contrast, simply differentiating through the SL-based first order gradient estimator again, as was done for MAML by Finn et al. (2017), results in omitted gradient terms and a biased gradient estimator, as pointed out by Al-Shedivat et al. (2017) and Stadie et al. (2018).

Figure 1 shows a comparison between the LOLA-DICE agents and the original formulation of LOLA. In our experiments, we use a time horizon of 150 steps and a reduced batch size of 64; the lookahead gradient step, $\alpha$, is set to 1 and the learning rate is 0.3. Importantly, given the approximation used, the LOLA method was restricted to a single step of opponent learning. In contrast, using DICE we can differentiate through an arbitrary number of opponent learning steps.

The original LOLA implemented via second order gradient corrections shows no stable learning, as it requires much larger batch sizes ($\sim 4000$). In contrast, LOLA-DICE agents discover strategies of high social welfare, replicating the results of the original LOLA paper in a way that is both more direct, efficient and establishes a common formulation between MAML and LOLA.

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

## A RELATED WORK

Gradient estimation is well studied, although many methods have been named and explored independently in different fields, and the primary focus has been on first order gradients. Fu (2006) provides an overview of methods from the point of view of simulation optimization.

The score function (SF) estimator, also referred to as the likelihood ratio estimator or REINFORCE, has received considerable attention in many fields. In reinforcement learning, policy gradient methods (Williams, 1992) have proven highly successful, especially when combined with variance reduction techniques (Weaver & Tao, 2001; Grondman et al., 2012). The SF estimator has also been used in the analysis of stochastic systems (Glynn, 1990), as well as for variational inference (Wingate & Weber, 2013; Ranganath et al., 2014).

Kingma & Welling (2013) and Rezende et al. (2014) discuss Monte-Carlo gradient estimates in the case where the stochastic parts of a model are amenable to reparameterisation.

To easily make use of these estimates for optimizing neural network models, automatic differentiation for backpropagation has been widely used (Baydin et al., 2015).

These approaches are formalized for arbitrary computation graphs by Schulman et al. (2015a), but to our knowledge our paper is the first to present a practical and correct approach for generating higher order gradient estimators utilizing auto-diff.

One rapidly growing application area for such higher order gradient estimates is meta-learning for reinforcement learning. Finn et al. (2017) compute a loss after a single policy gradient learning step, differentiating through the learning step to find parameters that can be quickly fine-tuned for different tasks. Li et al. (2017) extend this work to also meta-learn the fine-tuning step direction and magnitude.

Al-Shedivat et al. (2017) and Stadie et al. (2018) derive the proper higher order gradient estimators for their work by reapplying the score function trick. Foerster et al. (2018) instead use a Taylor expansion to derive their gradient estimators. None present a general strategy for constructing higher order gradient estimators for arbitrary graphs.

## B BACKGROUND

Suppose $x$ is a random variable, $x \sim p(x; \theta)$, $f$ is a function of $x$ and we want to compute $\nabla_\theta \mathbb{E}_x[f(x)]$. If the analytical gradients $\nabla_\theta f$ are unavailable or nonexistent, we can employ the *score function* (SF) estimator:

$$\nabla_\theta \mathbb{E}_x[f(x)] = \mathbb{E}_x[f(x)\nabla_\theta \log(p(x; \theta))] \tag{B.1}$$

If instead $x$ is a deterministic function of $\theta$ and another random variable $z$, the operators $\nabla_\theta$ and $\mathbb{E}_z$ commute, yielding the *pathwise derivative estimator* or *reparameterisation trick*. In this work, we focus on the SF estimator, which can capture the interdependency of both the objective and the sampling distribution on the parameters $\theta$, and therefore requires careful handling for higher order gradient estimates.

## B.1 STOCHASTIC COMPUTATION GRAPHS

Gradient estimators for single random variables can be generalised using the formalism of a stochastic computation graph (SCG, Schulman et al., 2015a). An SCG is a directed acyclic graph with four types of nodes: *input nodes*, $\Theta$; *deterministic nodes*, $\mathcal{D}$; *cost nodes*, $\mathcal{C}$; and *stochastic nodes*, $\mathcal{S}$. Input nodes are set externally and can hold parameters we seek to optimise. Deterministic nodes are functions of their parent nodes, while stochastic nodes are distributions conditioned on their parent nodes. The set of cost nodes $\mathcal{C}$ are those associated with an objective $\mathcal{L} = \mathbb{E}[\sum_{c \in \mathcal{C}} c]$.

Let $v \prec w$ denote that node $v$ *influences* node $w$, *i.e.*, there exists a path in the graph from $v$ to $w$. If every node along the path is deterministic, $v$ influences $w$ deterministically which is denoted by $v \prec^D w$. See Figure 2 (top) for a simple SCG with an input node $\theta$, a stochastic node $x$ and a cost function $f$. Note that $\theta$ influences $f$ deterministically ($\theta \prec^D f$) as well as stochastically via $x$ ($\theta \prec f$).

## B.2 SURROGATE LOSSES

In order to estimate gradients of a sum of cost nodes, $\sum_{c \in \mathcal{C}} c$, in an arbitrary SCG, Schulman et al. (2015a) introduce the notion of a *surrogate loss* (SL):

$$\mathrm{SL}(\Theta, \mathcal{S}) := \sum_{w \in \mathcal{S}} \log p(w \mid \mathrm{DEPS}_w) \hat{Q}_w + \sum_{c \in \mathcal{C}} c(\mathrm{DEPS}_c)$$

Here $\mathrm{DEPS}_w$ are the "dependencies" of $w$: the set of stochastic or input nodes that deterministically influence the node $w$. Furthermore, $\hat{Q}_w$ is the sum of *sampled* costs $\hat{c}$ corresponding to the cost nodes influenced by $w$.

The SL produces a gradient estimator when differentiated once (Schulman et al., 2015a, Corollary 1):

$$\nabla_\theta \mathcal{L} = \mathbb{E}[\nabla_\theta \mathrm{SL}(\Theta, \mathcal{S})]. \tag{B.2}$$

The hat notation on $\hat{Q}_w$ indicates that, inside the SL, these costs are treated as fixed samples, thus severing the functional dependency on $\theta$ that was present in the original stochastic computation graph. This ensures that the first order gradients of the SL match the score function estimator, which does not contain a term of the form $\log(p) \nabla_\theta Q$.

Although Schulman et al. (2015a) focus on first order gradients, they argue that the SL gradient estimates themselves can be treated as costs in an SCG and that the SL approach can be applied repeatedly to construct higher order gradient estimators. However, the use of sampled costs in the SL leads to missing dependencies and wrong estimates when calculating such higher order gradients, as we discuss in Section C.2.

## C HIGHER ORDER GRADIENTS

In this section, we illustrate how to estimate higher order gradients via repeated application of the score function (SF) trick and show that repeated application of the surrogate loss (SL) approach in stochastic computation graphs (SCGs) fails to capture all of the relevant terms for higher order gradient estimates.

### C.1    HIGHER ORDER GRADIENT ESTIMATORS

We begin by revisiting the derivation of the score function estimator for the gradient of the expectation $\mathcal{L}$ of $f(x; \theta)$ over $x \sim p(x; \theta)$:

$$
\begin{aligned}
\nabla_\theta \mathcal{L} &= \nabla_\theta \mathbb{E}_x \left[ f(x; \theta) \right] \\
&= \nabla_\theta \sum_x p(x; \theta) f(x; \theta) \\
&= \sum_x \nabla_\theta \left( p(x; \theta) f(x; \theta) \right) \\
&= \sum_x \left( f(x; \theta) \nabla_\theta p(x; \theta) + p(x; \theta) \nabla_\theta f(x; \theta) \right) \\
&= \sum_x \left( f(x; \theta) p(x; \theta) \nabla_\theta \log(p(x; \theta)) \right. \\
&\qquad\qquad \left. + p(x; \theta) \nabla_\theta f(x; \theta) \right) \\
&= \mathbb{E}_x \left[ f(x; \theta) \nabla_\theta \log(p(x; \theta)) + \nabla_\theta f(x; \theta) \right] \qquad\text{(C.1)} \\
&= \mathbb{E}_x [g(x; \theta)].
\end{aligned}
$$

The estimator $g(x; \theta)$ of the gradient of $\mathbb{E}_x \left[ f(x; \theta) \right]$ consists of two distinct terms: (1) The term $f(x; \theta) \nabla_\theta \log(p(x; \theta))$ originating from $f(x; \theta) \nabla_\theta p(x; \theta)$ via the SF trick, and (2) the term $\nabla_\theta f(x; \theta)$, due to the direct dependence of $f$ on $\theta$. The second term is often ignored because $f$ is often only a function of $x$ but not of $\theta$. However, even in that case, the gradient estimator $g$ depends on both $x$ and $\theta$. We might be tempted to again apply the SL approach to $\nabla_\theta \mathbb{E}_x[g(x; \theta)]$ to produce estimates of higher order gradients of $\mathcal{L}$, but below we demonstrate that this fails. In Section D, we subsequently introduce a practical algorithm for correctly producing such higher order gradient estimators in SCGs.

### C.2    HIGHER ORDER SURROGATE LOSSES

While Schulman et al. (2015a) focus on the first order gradients, they state that a recursive application of SL can generate higher order gradient estimators. However, as we demonstrate in this section, because the SL approach treats part of the objective as a sampled cost, the corresponding terms lose a functional dependency on the sampling distribution. This leads to missing terms in the estimators of the higher order gradients.

Consider the following example, where a single parameter $\theta$ defines a sampling distribution $p(x; \theta)$ and the objective is $f(x, \theta)$.

$$
\begin{aligned}
\text{SL}(\mathcal{L}) &= \log p(x; \theta) \hat{f}(x) + f(x; \theta) \\
(\nabla_\theta \mathcal{L})_{\text{SL}} &= \mathbb{E}_x [\nabla_\theta \text{SL}(\mathcal{L})] \\
&= \mathbb{E}_x [\hat{f}(x) \nabla_\theta \log p(x; \theta) + \nabla_\theta f(x; \theta)] \qquad\text{(C.2)} \\
&= \mathbb{E}_x [g_{\text{SL}}(x; \theta)].
\end{aligned}
$$

The corresponding SCG is depicted at the top of Figure 2. Comparing (C.1) and (C.2), note that the first term, $\hat{f}(x)$ has lost its functional dependency on $\theta$, as indicated by the hat notation and the lack of a $\theta$ argument. While these terms evaluate to the same estimate of the first order gradient, the lack of the functional dependency yields a discrepancy between the exact derivation of the second order gradient and a second application of SL.

$$
\begin{aligned}
\text{SL}(g_{\text{SL}}(x; \theta)) &= \log p(x; \theta) \hat{g}_{\text{SL}}(x) + g_{\text{SL}}(x; \theta) \\
(\nabla_\theta^2 \mathcal{L})_{\text{SL}} &= \mathbb{E}_x [\nabla_\theta \text{SL}(g_{\text{SL}})] \\
&= \mathbb{E}_x [\hat{g}_{\text{SL}}(x) \nabla_\theta \log p(x; \theta) + \nabla_\theta g_{\text{SL}}(x; \theta)]. \qquad\text{(C.3)}
\end{aligned}
$$

By contrast, the exact derivation of $\nabla_\theta^2 \mathcal{L}$ results in the following expression:

$$
\begin{aligned}
\nabla_\theta^2 \mathcal{L} &= \nabla_\theta \mathbb{E}_x [g(x; \theta)] \\
&= \mathbb{E}_x [g(x; \theta) \nabla_\theta \log p(x; \theta) + \nabla_\theta g(x; \theta)]. \qquad\text{(C.4)}
\end{aligned}
$$

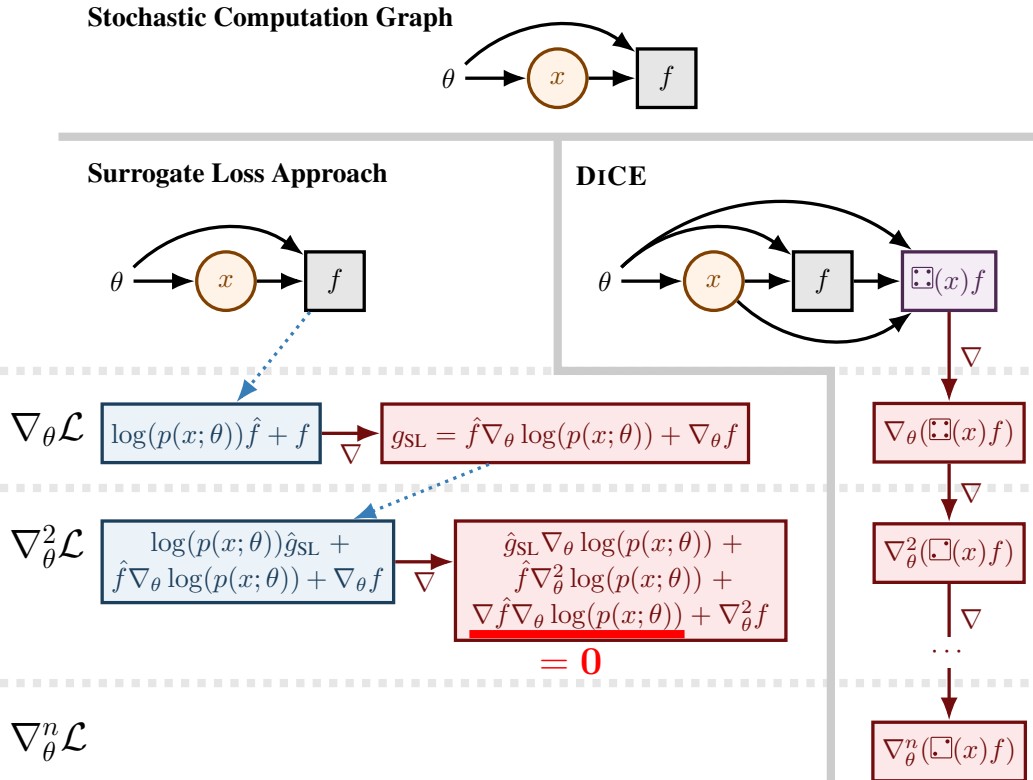

Figure 2: Simple example illustrating the difference of the Surrogate Loss (SL) approach to DICE. Stochastic nodes are depicted in orange, costs in gray, surrogate losses in blue, DICE in purple, and gradient estimators in red. Note that for second-order gradients, SL requires the construction of an intermediate stochastic computation graph and due to taking a sample of the cost $\hat{g}_{\text{SL}}$, the dependency on $\theta$ is lost, leading to an incorrect second-order gradient estimator. Arrows from $\theta, x$ and $f$ to gradient estimators omitted for clarity.

Since $g_{\text{SL}}(x; \theta)$ differs from $g(x; \theta)$ only in its functional dependencies on $\theta$, $g_{\text{SL}}$ and $g$ are identical when *evaluated*. However, due to the missing dependencies in $g_{\text{SL}}$, the *gradients* w.r.t. $\theta$, which appear in the higher order gradient estimates in (C.3) and (C.4), differ:

$$\nabla_\theta g(x; \theta) = \nabla_\theta f(x; \theta) \nabla_\theta \log(p(x; \theta))$$
$$+ f(x; \theta) \nabla_\theta^2 \log(p(x; \theta))$$
$$+ \nabla_\theta^2 f(x; \theta),$$
$$\nabla_\theta g_{\text{SL}}(x; \theta) = \hat{f}(x) \nabla_\theta^2 \log(p(x; \theta))$$
$$+ \nabla_\theta^2 f(x; \theta).$$

We lose the term $\nabla_\theta f(x; \theta) \nabla_\theta \log(p(x; \theta))$ in the second order SL gradient because $\nabla_\theta \hat{f}(x) = 0$ (see left part of Figure 2). This issue occurs immediately in the second order gradients when $f$ depends directly on $\theta$. However, as $g(x; \theta)$ always depends on $\theta$, the SL approach always fails to produce correct third or higher order gradient estimates even if $f$ depends only indirectly on $\theta$.

### C.3 SIMPLE FAILING EXAMPLE

Here is a toy example to illustrate a possible failure case. Let $x \sim \text{Ber}(\theta)$ and $f(x, \theta) = x(1 - \theta) + (1 - x)(1 + \theta)$. For this simple example we can exactly evaluate all terms:

$$\mathcal{L} = \theta(1 - \theta) + (1 - \theta)(1 + \theta)$$
$$= -2\theta^2 + \theta + 1$$
$$\nabla_\theta \mathcal{L} = -4\theta + 1$$
$$\nabla_\theta^2 \mathcal{L} = -4$$

Evaluating the expectations for the SL-gradient estimators analytically results in the following terms:

$$(\nabla_\theta \mathcal{L})_{\text{SL}} = -4\theta + 1$$
$$(\nabla_\theta^2 \mathcal{L})_{\text{SL}} = -2$$

Even with an infinite number of samples, the SL estimator produces the wrong second order gradient. If, for example, these wrong estimates were used in combination with the Newton-Raphson method for optimising $\mathcal{L}$, then $\theta$ would never converge to the correct value. In contrast, this method would converge in a single step using the correct gradients.

The failure mode seen in this toy example will appear whenever the objective includes a regularisation term that depends on $\theta$, and is also impacted by the stochastic samples. One example in a practical algorithm is soft $Q$-learning for RL (Schulman et al., 2017), which regularises the policy by adding an entropy penalty to the rewards. This penalty encourages the agent to maintain an exploratory policy, reducing the probability of getting stuck in local optima. Clearly the penalty depends on the policy parameters $\theta$. However, the policy entropy will also depends on the states visited, which in turn depend on the stochastically sampled actions. As a result, the entropy regularised RL objective in this algorithm will have the exact property leading to the failure of the SL approach shown above. Unlike our toy analytic example, the consequent errors will not just appear as a rescaling of the proper higher order gradients, but will depend in a complex way on the parameters $\theta$. Any second order methods with such a regularised objective will therefore require an alternative strategy for generating gradient estimators, even setting aside the awkwardness of repeatedly generating new surrogate objectives.

## D CORRECT GRADIENT ESTIMATORS WITH DICE

In this section, we propose the *Infinitely Differentiable Monte-Carlo Estimator*(DICE), a practical algorithm for programatically generating correct gradients of any order in arbitrary SCGs. The naive option is to recursively apply the update rules in (C.1) that map from $f(x; \theta)$ to the estimator of its derivative $g(x; \theta)$. However, this approach has two deficiencies: First, by defining gradients directly, it fails to provide an objective that can be used in standard deep learning libraries. Second, these naive gradient estimators violate the auto-diff paradigm for generating further estimators by repeated differentiation since in general $\nabla_\theta f(x; \theta) \neq g(x; \theta)$. Our approach addresses these issues, as well as fixing the missing terms from the SL approach.

As before, $\mathcal{L} = \mathbb{E}[\sum_{c \in \mathcal{C}} c]$ is the objective in an SCG. The correct expression for a gradient estimator that preserves all required dependencies for further differentiation is:

$$\nabla_\theta \mathcal{L} = \mathbb{E}\left[ \sum_{c \in \mathcal{C}} \left( c \sum_{w \in \mathcal{W}_c} \nabla_\theta \log p(w \mid \text{DEPS}_w) \right. \right.$$
$$\left. \left. + \nabla_\theta c(\text{DEPS}_c) \right) \right], \tag{D.1}$$

where $\mathcal{W}_c = \{w \mid w \in \mathcal{S}, w \prec c, \theta \prec w\}$, *i.e.* the set of stochastic nodes that depend on $\theta$ and influence the cost $c$. For brevity, from here on we suppress the DEPS notation, assuming all probabilities and costs are conditioned on their relevant ancestors in the SCG.

Note that (D.1) is the generalisation of (C.1) to arbitrary SCGs. The proof is given by Schulman et al. (2015a, Lines 1-10, Appendix A). Crucially, in Line 11 the authors then replace $c$ by $\hat{c}$, severing the dependencies required for correct higher order gradient estimators. As described in Section B.2, this was done so that the SL approach reproduces the score function estimator after a single differentiation and can thus be used as an objective for backpropagation in a deep learning library.

To support correct higher order gradient estimators, we propose DICE, which relies heavily on a novel operator, MAGICBOX($\boxdot$). This operator takes a set of stochastic nodes $\mathcal{W}$ as input and has the following two properties by design:

1. $\boxdot(\mathcal{W}) \rightarrowtail 1$,
2. $\nabla_\theta \boxdot(\mathcal{W}) = \boxdot(\mathcal{W}) \sum_{w \in \mathcal{W}} \nabla_\theta \log(p(w; \theta))$.

Here, $\rightarrowtail$ indicates "*evaluates to*" in contrast to full equality, $=$, which includes equality of all gradients. In the auto-diff paradigm, $\rightarrowtail$ corresponds to a forward pass evaluation of a term. Meanwhile, the behaviour under differentiation in property (2) indicates the new graph nodes that will be constructed to hold the gradients of that object. Note that that $\boxdot(\mathcal{W})$ reproduces the dependency of the gradient on the sampling distribution under differentiation through the requirements above. Using $\boxdot$, we can next define the DICE objective, $\mathcal{L}_\boxdot$:

$$\mathcal{L}_\boxdot = \sum_{c \in \mathcal{C}} \boxdot(\mathcal{W}_c) c. \tag{D.2}$$

Below we prove that the DICE objective indeed produces correct arbitrary order gradient estimators under differentiation.

**Theorem 1.** $\mathbb{E}[\nabla_\theta^n \mathcal{L}_\boxdot] \rightarrowtail \nabla_\theta^n \mathcal{L}, \forall n \in \{0, 1, 2, \dots\}$.

*Proof.* For each cost node $c \in \mathcal{C}$, we define a sequence of nodes, $c^n, n \in \{0, 1, \dots\}$ as follows:

$$c^0 = c,$$
$$\mathbb{E}[c^{n+1}] = \nabla_\theta \mathbb{E}[c^n]. \tag{D.3}$$

By induction it follows that $\mathbb{E}[c^n] = \nabla_\theta^n \mathbb{E}[c] \; \forall n$, *i.e.* that $c^n$ is an estimator of the $n$th order derivative of the objective $\mathbb{E}[c]$.

We further define $c_\boxdot^n = c^n \boxdot(\mathcal{W}_{c^n})$. Since $\boxdot(x) \rightarrowtail 1$, clearly $c_\boxdot^n \rightarrowtail c^n$. Therefore $\mathbb{E}[c_\boxdot^n] \rightarrowtail \mathbb{E}[c^n] = \nabla_\theta^n \mathbb{E}[c]$, *i.e.*, $c_\boxdot^n$ is also a valid estimator of the $n$th order derivative of the objective. Next, we show that $c_\boxdot^n$ can be generated by differentiating $c_\boxdot^0$ $n$ times. This follows by induction, if $\nabla_\theta c_\boxdot^n = c_\boxdot^{n+1}$, which we prove as follows:

$$
\begin{aligned}
\nabla_\theta c_\boxdot^n &= \nabla_\theta(c^n \boxdot(\mathcal{W}_{c^n})) \\
&= c^n \nabla_\theta \boxdot(\mathcal{W}_{c^n}) + \boxdot(\mathcal{W}_{c^n}) \nabla_\theta c^n \\
&= c^n \boxdot(\mathcal{W}_{c^n}) \left( \sum_{w \in \mathcal{W}_{c^n}} \nabla_\theta \log(p(w; \theta)) \right) \\
&\quad + \boxdot(\mathcal{W}_{c^n}) \nabla_\theta c^n \\
&= \boxdot(\mathcal{W}_{c^n}) \left( \nabla_\theta c^n + c^n \sum_{w \in \mathcal{W}_{c^n}} \nabla_\theta \log(p(w; \theta)) \right) \tag{D.4} \\
&= \boxdot(\mathcal{W}_{c^{n+1}}) c^{n+1} = c_\boxdot^{n+1}. \tag{D.5}
\end{aligned}
$$

To proceed from (D.4) to (D.5), we need two additional steps. First, we require an expression for $c^{n+1}$. Substituting $\mathcal{L} = \mathbb{E}[c^n]$ into (D.1) and comparing to (D.3), we find the following map from $c^n$ to $c^{n+1}$:

$$c^{n+1} = \nabla_\theta c^n + c^n \sum_{w \in \mathcal{W}_{c^n}} \nabla_\theta \log p(w; \theta). \tag{D.6}$$

The term inside the brackets in (D.4) is identical to $c^{n+1}$. Secondly, note that (D.6) shows that $c^{n+1}$ depends only on $c^n$ and $\mathcal{W}_{c^n}$. Therefore, the stochastic nodes which influence $c^{n+1}$ are the same as those which influence $c^n$. So $\mathcal{W}_{c^n} = \mathcal{W}_{c^{n+1}}$, and we arrive at (D.5).

To conclude the proof, recall that $c^n$ is the estimator for the $n$th derivative of $c$, and that $c_\square^n \rightarrowtail c^n$. Summing over $c \in \mathcal{C}$ then gives the desired result. $\qquad\square$

## D.1   Implementation of DiCE

DiCE is easy to implement in standard deep learning libraries:

$$\boxdot(\mathcal{W}) = \exp\big(\tau - \bot(\tau)\big),$$
$$\tau = \sum_{w \in \mathcal{W}} \log(p(w;\theta)),$$

where $\bot$ is an operator that sets the gradient of the operand to zero, so $\nabla_x \bot(x) = 0$. [1]

Since $\bot(x) \rightarrowtail x$, clearly $\boxdot(\mathcal{W}) \rightarrowtail 1$. Furthermore:

$$\begin{aligned}
\nabla_\theta \boxdot(\mathcal{W}) &= \nabla_\theta \exp\big(\tau - \bot(\tau)\big) \\
&= \exp\big(\tau - \bot(\tau)\big) \nabla_\theta(\tau - \bot(\tau)) \\
&= \boxdot(\mathcal{W})(\nabla_\theta \tau + 0) \\
&= \boxdot(\mathcal{W}) \sum_{w \in \mathcal{W}} \nabla_\theta \log(p(w;\theta)).
\end{aligned}$$

With this implementation of the $\boxdot$-operator, it is now straightforward to construct $\mathcal{L}_\square$ as defined in (D.7). This procedure is demonstrated in Figure 3, which shows a reinforcement learning use case. In this example, the cost nodes are rewards that depend on stochastic actions, and the total objective is $J = \mathbb{E}[\sum r_t]$. We construct a DiCE objective $J_\square = \sum_t \boxdot(\{a_{t'}, t' \le t\})r_t$. Now $\mathbb{E}[J_\square] \rightarrowtail J$ and $\mathbb{E}[\nabla_\theta^n J_\square] \rightarrowtail \nabla_\theta^n J$, so $J_\square$ can both be used to estimate the return and to produces estimators for any order gradients under auto-diff, which can be used for higher order methods such as TRPO (Schulman et al., 2015b).

**Causality.** In Theorem 1 of Schulman et al. (2015a), two expressions for the the gradient estimator are provided:

1. The first expression handles causality by summing over stochastic nodes, $w$, and multiplying $\nabla \log(p(w))$ for each stochastic node with a sum of the *downstream* costs, $\hat{Q}_w$. This is *forward looking* causality.

2. In contrast, the second expression sums over costs, $c$, and multiplies each cost with a sum over the gradients of log-probabilities from *upstream* stochastic nodes, $\sum_{w \in \mathcal{W}_c} \nabla \log(p(w))$. We can think of this as *backward looking* causality.

In both cases, integrating causality into the gradient estimator leads to reduction of variance compared to the naive approach of multiplying the full sum over costs with the full sum over grad-log-probabilities.

While the SL approach is based on the first expression, DiCE uses the second formulation. As shown by Schulman et al. (2015a), both expressions result in the same terms for the gradient estimator. However, the second formulation leads to greatly reduced conceptual complexity when calculating higher order terms, which we exploit in the definition of the DiCE objective. This is because each further gradient estimator maintains the same backward looking dependencies for each term in the original sum over costs, *i.e.*, $\mathcal{W}_{c^n} = \mathcal{W}_{c^{n+1}}$. In contrast, the SL approach is centred around the stochastic nodes, which each become associated with a growing number of downstream costs after each differentiation. Consequently, we believe that our DiCE objective is more intuitive, as it is conceptually centred around the original objective and remains so under repeated differentiation.

## D.2   Variance Reduction.

So far, the construction of the DiCE objective addresses variance reduction only via implementing causality, since each cost term is associated with the $\boxdot$ that captures all causal dependencies. However,

---

[1]This operator exists in PyTorch as `detach` and in TensorFlow as `stop_gradient`.

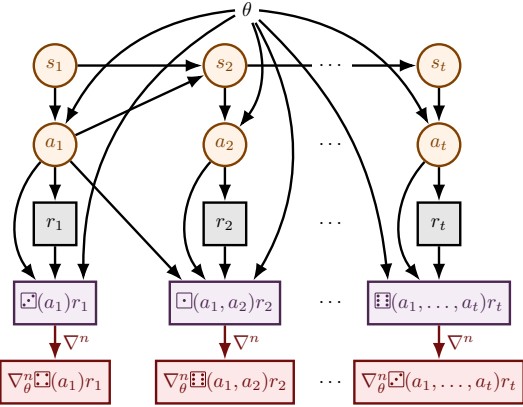

Figure 3: DICE applied to a reinforcement learning problem. A stochastic policy conditioned on $s_t$ and $\theta$ produces actions, $a_t$, which lead to rewards $r_t$ and next states, $s_{t+1}$. Associated with each reward is a DICE objective that takes as input the set of all causal dependencies that are functions of $\theta$, *i.e.*, the actions. Arrows from $\theta, a_i$ and $r_i$ to gradient estimators omitted for clarity.

we can also include a baseline term in the definition of the DICE objective:

$$\mathcal{L}_{\boxdot} = \sum_{c \in \mathcal{C}} \boxdot(\mathcal{W}_c)c + \sum_{w \in \mathcal{S}} (1 - \boxdot(\{w\}))b_w. \tag{D.7}$$

The baseline $b_w$ is a design choice and can be any function of nodes not influenced by $w$. As long as this condition is met, the baseline will not change the expectation of the gradient estimates, but can considerably reduce the variance (including of higher order gradient estimators). A common choice is the average cost.

Since $(1 - \boxdot(\{w\})) \rightarrowtail 0$, the addition of the baseline leaves the evaluation of the estimator $\mathcal{L}_{\boxdot}$ of the original objective unchanged, while reproducing the baseline term stated by Schulman et al. (2015a) after differentiation.

**Hessian-Vector Product.** The Hessian-vector, $v^\top H$, is useful for a number of algorithms, such as estimation of eigenvectors and eigenvalues of $H$ (Pearlmutter, 1994). Using DICE, $v^\top H$ can be implemented efficiently without having to compute the full Hessian. Assuming $v$ does not depend on $\theta$ and using $^\top$ to indicate the transpose:

$$\begin{aligned} v^\top H &= v^\top \nabla^2 \mathcal{L}_{\boxdot} \\ &= v^\top (\nabla^\top \nabla \mathcal{L}_{\boxdot}) \\ &= \nabla^\top (v^\top \nabla \mathcal{L}_{\boxdot}). \end{aligned}$$

In particular, $(v^\top \nabla \mathcal{L}_{\boxdot})$ is a scalar, making this implementation well suited for auto-diff.

## E  EMPIRICAL VERIFICATION.

We first verify that DICE successfully recovers gradients and Hessians in stochastic computation graphs. To do so, we use DICE to estimate gradients and Hessians of the expected return for fixed policies in IPD.

As shown in Figure 4, we find that indeed the DICE estimator matches both the gradients (a) and the Hessians (b) for both agents accurately. Furthermore, Figure 5 shows how the estimate of the gradient improve as the value function becomes more accurate during training, in (a). Also shown is the quality of the gradient estimation as a function of sample size with and without a baseline, in (b). Both plots show that the baseline is a key component of DICE for accurate estimation of gradients.

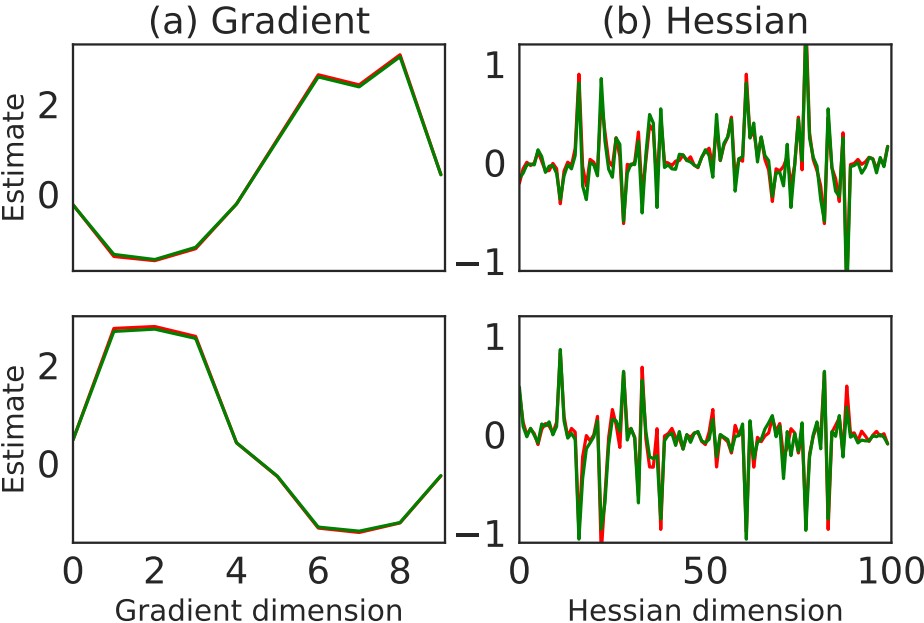

Figure 4: For each of the two agents (1 top row, 2 bottom row) in the iterated prisoner's dilemma, shown is the flattened true (red) and estimated (green) Gradient (left) and Hessian (right) using the first and second derivative of DICE and the exact value function respectively. The correlation coefficients are 0.999 for the gradients and 0.97 for the Hessian; the sample size is 100k.

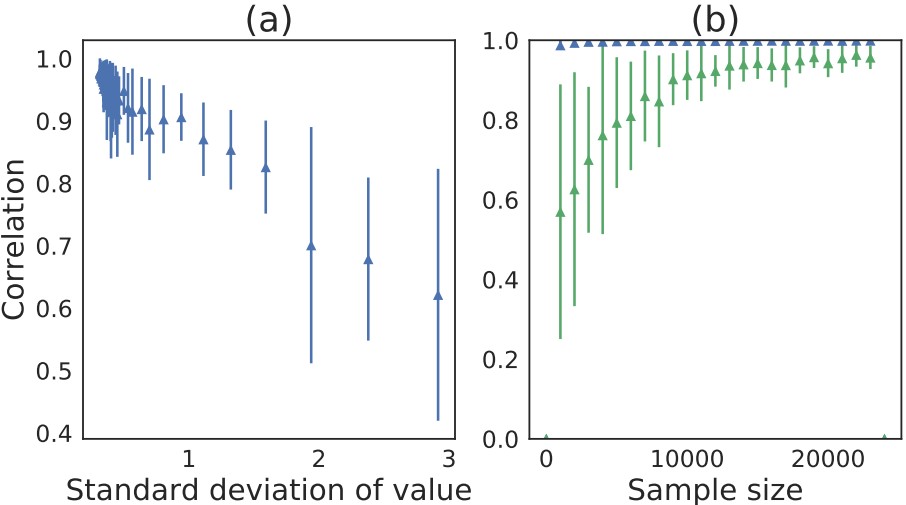

Figure 5: Shown in (a) is the correlation of the gradient estimator (averaged across agents) as a function of the estimation error of the baseline when using a sample size of 128 and in (b) as a function of sample size when using a converged baseline (in blue) and no baseline (in green). In both plots errors bars indicate the standard deviation.

