# OpenReview forum: "DiCE: The Infinitely Differentiable Monte-Carlo Estimator"
_ICLR.cc/2018/Workshop — Accept_

### Official Review · AnonReviewer2 · 2018-03-09
**Interesting improvement to the surrogate loss approach**

**Rating:** 6
**Confidence:** 1

**Review:**

The authors propose a new approach for estimating gradients of stochastic objectives, designed for use in automatic-differentiation toolboxes. It is shown, both analytically and empirically, that it addresses some of the shortcomings in the existing approach based on surrogate loss in estimating higher order gradients.

While I did not go through the full content in the appendix, the work seems convincing and is very relevant to the learning community.

I find the random dice symbol distracting.

---

### Official Review · AnonReviewer1 · 2018-03-09
**The authors provide a novel SL objective that is amenable to autograd even when it comes to calculating higher order gradients**

**Rating:** 6
**Confidence:** 1

**Review:**

Score functions are used to produce Monte Carlo estimates of gradients.  Surrogate loss function based approach provides a way to provide gradient estimates that utilizes auto-diff and stochastic computational graphs to programmatically calculate gradients. However, the SL approach does not provide an easy way to calculate higher order gradients easily, without tedious manual calculations.
The authors in this paper provide a way to get around this problem. They do this by producing a novel operator called DICE, which wraps around the set of stochastic nodes that influence the original loss function.  The authors demonstrate this estimator for multi-agent RL  and show that LOLA-DICE is able to differentiate through multiple steps of the opponent,  whereas LOLA and MAML are somewhat unstable and are restricted to only differentiating w.r.t. one gradient step of the opponent.
This is interesting work, but I am unable to judge the correctness of this work. For example, the formulation of DICE objective on page 2 is very unclear. For example, there are multiple dice symbols with multiple, varying dots. i guess they refer to different operators. However, it is not clear to me what they mean.

---

### Official Review · AnonReviewer3 · 2018-03-10
**Nice paper that fixes the computation of higher-order gradients in stochastic computation graphs, enabling correct auto-diff training for RL policy gradient, soft Q-learning, model-agnostic meta learning,  etc.**

**Rating:** 8
**Confidence:** 4

**Review:**

The paper crisply points out the flaw in existing implementations to get gradients in stochastic computation graphs (i.e. they don't generalize nicely for higher-order gradients, which is a natural requirement for many applications). The fix is explained in a simple manner, and easily implemented in TensorFlow, pyTorch etc.

Clarity: Average. The Dice symbol for MAGICBOX is distracting notation! (First instinct was that it is cute and clever, but it actually hindered understanding of the equations).

Significance: Good. Paper has a singular purpose, but it correctly points out bugs in prior work (e.g. MAML) and makes algorithms like LOLA more practical and so, this can be a significant contribution.

---

### Decision · Program_Chairs · 2018-03-20
**ICLR 2018 Workshop Acceptance Decision**

**Decision:**

Accept

**Comment:**

Congratulations, your paper was accepted to the ICLR workshop.